# CircRNA Expression Profiles in Canine Mammary Tumours

**DOI:** 10.3390/vetsci9050205

**Published:** 2022-04-22

**Authors:** Yufan Zhu, Baochun Lu, Juye Wu, Shoujun Li, Kun Jia

**Affiliations:** 1College of Veterinary Medicine, South China Agricultural University, Guangzhou 510642, China; 20212029018@stu.scau.edu.cn (Y.Z.); lubaochum@stu.scau.edu.cn (B.L.); wujuye@scau.edu.cn (J.W.); shoujunli@scau.edu.cn (S.L.); 2Guangdong Provincial Key Laboratory of Prevention and Control for Severe Clinical Animal Diseases, Guangzhou 510642, China; 3Guangdong Technological Engineering Research Center for Pet, Guangzhou 510642, China

**Keywords:** canine mammary tumours, circRNAs, expression profiles, biomarker

## Abstract

Numerous studies have shown that the occurrence and development of tumours are associated with the expression of circular RNAs (circRNAs). However, the expression profile and clinical significance of circRNAs in canine mammary tumours remain unclear. In this paper, we collected tissue samples from three dogs with canine mammary tumours and analysed the expression profiles of circRNAs in these samples using high-throughput sequencing technology. GO (Gene Ontology) and KEGG (Kyoto Encyclopedia of Genes and Genomes) analyses revealed 14 biological processes associated with these genes, and 11 of these genes were selected for qRT-PCR to verify their authenticity. CircRNAs have sponge adsorption to miRNAs, so we constructed a circRNA-miRNA network map using Cytoscape software. As a result, we identified a total of 14,851 circRNAs in canine mammary tumours and its adjacent normal tissues. Of these, 106 were differentially expressed (fold change ≥ 2, *p* ≤ 0.05), and 64 were upregulated and 42 were downregulated. The GO analysis revealed that the biological processes involved were mainly in the regulation of the secretory pathway, the regulation of neurotransmitter secretion and the positive regulation of phagocytosis. Most of these biological pathways were associated with the cGMP-PKG (cyclic guanosine monophosphate) signalling pathway, the cAMP (cyclic adenosine monophosphate) signalling pathway and the oxytocin signalling pathway. After screening, source genes closely associated with canine mammary tumours were found to include RYR2, PDE4D, ROCK2, CREB3L2 and UBA3, and associated circRNAs included chr27:26618544-26687235-, chr26:8194880-8201833+ and chr17:7960861-7967766-. In conclusion, we reveals the expression profile of circRNAs in canine mammary tumours. In addition, some circRNAs might be used as potential biomarkers for molecular diagnosis.

## 1. Introduction

CircRNAs are a new type of non-coding RNA that have a covalently closed loop structure and are produced by back splice [1]. Although they were once thought to be garbage sequence [2], with more in-depth research, their unique attributes and functions have been increasingly recognized by scientists. CircRNAs are a closed ring molecular structure; thus, they are not affected by RNA exonuclease, are not easily degraded and have more stable expression than linear RNA in the cell [3]. Recent studies showed that circRNAs contain binding sites of miRNA and RNA binding protein (RBP); thus, they play an important role in the occurrence and development of disease [4]. CircRNAs are expressed widely and are conserved across species; they also display stability and specificity in different tissues [5]. Moreover, numerous studies have shown that circRNAs are involved in a variety of cellular processes and are involved in the development and progression of many diseases, particularly cancer [6,7,8,9,10]. For example, circANKS1B increases the expression of the transcription factor USF1 by sponging miR-148a-3p and miR-152-3P, which in turn upregulates TGF-β1, activates the TGF-β1/Smad signalling pathway, promotes the conversion of breast epithelial cells into mesenchymal cells, and finally promotes breast cancer invasion and metastasis [11]; Cir-GLI2 promotes proliferation and migration as well as the invasion of osteosarcoma cells by targeting miR-125b-5p [12]. ciRS-7 is significantly upregulated in colorectal cancer tissue compared with normal intestinal mucosa, and the overexpression of ciRS-7 leads to miR-7 blockage and results in a more aggressive oncogenic phenotype [13]. The expression level of hsa_circ_0000745 in GC tissues is related to tumour differentiation, and the expression level in plasma is related to the tumour lymph node metastasis stage [14]. It has been found that circRNAs have an impact on many cancers, including colorectal cancer, gastric cancer and mammary cancer. For example, circCSPP1 expression was significantly upregulated in colon cancer tissues and cell lines, and overexpression of circCSPP1 significantly promoted the proliferation, migration and invasion of colon cancer cells, while silencing of circCSPP1 had the opposite effect [15]. Therefore, they may be used as new biomarkers of cancer in the future [12,13,14,15].

Previous studies have shown that the incidence of canine mammary tumours is second only to skin tumours [16]. The incidence of canine mammary tumours accounted for 50% of all canine tumours [17]. There are many factors leading to death from canine mammary tumours, and early diagnosis is the key to early detection and intervention. Although the combination of X-ray, MRI (magnetic resonance imaging) and other imaging techniques can help make an early diagnosis, there are still significant limitations. Importantly, the highly proliferative and aggressive nature of canine mammary tumours often results in a poor prognosis for affected dogs [18,19,20,21]. In recent years, tumour biomarkers such as p53, COX-2 (cyclooxygenase-2), Bax and Bcl-2 (B-cell lymphoma-2) have been widely used for early diagnosis, for monitoring and for the prognosis of canine mammary tumours [22,23,24]. Currently, the application of biomarkers to detect disease has become a routine approach, but unfortunately, tumour markers have limited sensitivity and specificity for the diagnosis of canine mammary tumours. Moreover, numerous scholars have focused their research on transcriptomics and genomic mutations [25,26,27], but there is still no effective tumour biomarker for the pre-diagnosis of canine mammary tumours. The performance and potential of circRNAs in canine mammary tumours remain unclear.

In this study, to evaluate whether circRNAs are involved in the formation and development of canine mammary tumours, circRNA expression profiles between tumours tissues and adjacent normal tissues were first explored in canine mammary tumours.

## 2. Materials and Methods

### 2.1. Sample Collection and Treatment

From September 2016 to March 2017, canine mammary tissue samples were collected from consecutively recruited diseased animals that received mastectomy at various animal hospitals in Guangzhou. A total of 17 samples were collected. Canine mammary tumour tissues and their adjacent normal tissues (2 cm from tumour tissue) were disassembled aseptically and were frozen in a storage tube. Samples were snap frozen in liquid nitrogen for 15 min and then stored at −80 °C. We eventually selected three samples of cases initially diagnosed as malignant mammary tumours for more specialised pathological diagnosis and follow-up experiments. Prior to this, these dogs had not been treated with radiotherapy or chemotherapy.

### 2.2. RNA Library Construction and CircRNA Sequencing

Total RNA of the samples was extracted with Trizol reagent (TaKaRa, Tokyo, Japan), following the instructions. RNA concentration was measured by a NanoDrop^®^ ND-1000 spectrophotometer (Agilent Inc., Santa Clara, CA, USA). The steps to prepare circRNA libraries for each sample were as follows: (1) enrichment of circRNAs was performed using the CircRNA Enrichment Kit (Cloud-seq, Shanghai, China). (2) A Tru Seq Stranded Total RNA Library Prep Kit (Illumina, San Diego, CA, USA) was used for RNA pre-processing, according to the instructions. (3) The BioAnalyzer 2100 system (Agilent Technologies, Santa Clara, CA, USA) was used to quantify and control the quality of the sequencing libraries. Sequencing followed the Illumina sequencing protocol, with libraries adjusted to single-stranded DNA molecules, recovered on a flow cytometer, followed by in situ amplification, and finally 150 sequencing runs on an Illumina Hi Seq Se-quencer (manufacturer, Los Angeles, CA, USA).

### 2.3. RNA-Seq Data Statistics and RNA Labelling

After sequencing with the Illumina Hi Seq 4000 sequencer, the raw double-stranded reads were harvested after quality filtering. Raw paired-end reads were harvested from the Illumina Hi Seq 4000 sequencer after quality filtering. First, Q30 was used to carry out quality control. Cutadapt software (v1.9.3) was used to strip headers, and low-quality reads for high-quality reads were obtained. Then, the high-quality reads were aligned to the reference genome or transcriptome using BWA-MEM software (v.0.7.12), and circRNA detection and characterization were conducted with CIRI software (v.1.2) [28]. CircRNAs were identified by the CircBase database and circ2Trait disease database [29,30], and novel circRNAs were labelled “novel”. The raw junction reads for all the samples were normalized by the total reads and the converted by log2. Fold changes ≥ 2.0 and *p* ≤ 0.05 were categorized as significantly differentially expressed circRNAs.

### 2.4. Identification of Differentially Expressed CircRNAs

After finishing the standardisation of the original reads, we chose the Bowtie2 software(v.2.3.5, Baltimore, MD, USA) to extract circRNAs from the clean reads as the reference sequences for subsequent analysis by comparing them with the reference genome [31], and the differentially expressed circRNAs between the two groups of samples were calculated. Hierarchical clustering was performed using Heatmap2 in software R gplots (v.3.1.1, Auckland, New Zealand), and circRNAs complying with the conditions of fold changes ≥ 2.0 and *p* ≤ 0.05 were filtered. Generally, clustering genes into the same cluster indicates similar biological functions.

### 2.5. Bioinformatics Analysis of Differentially Expressed CircRNAs

By GO (Gene Ontology; http://www.geneontology.org (accessed on 29 November 2021)) and KEGG (Kyoto Encyclopedia of Genes and Genomes; http://www.genome (accessed on 29 November 2021)), JP/KEGG, we analysed the circRNAs of these host genes, annotated some of the differentially expressed circRNAs and determined their biological functions.

### 2.6. Prediction of Interactions between CircRNAs and miRNAs

Multiple miRNA targets existed for some circRNA sequences; these targets could combine with miRNAs and could play a regulatory role in the organism [32]. miRNAs can affect the expression of hundreds of cancer causing genes; for example, CDR1as can affect the expression of miR-7, and miR-7 can inhibit the expression of many oncogenes (EGFR, Raf1, Pak1, Ack1, IGF1R and m TOR) [33]. Through these analytical tools, interactions between circRNAs and miRNAs were found, and circRNAs may affect miRNAs through sponge action [34,35]. CircRNA-miRNA network maps were constructed using Cytoscape software (v.3.9.1, Bethesda, MD, USA) to visually correlate circRNAs and their predicted miRNAs with the development of canine mammary tumours [36].

### 2.7. Validation of Differentially Expressed CircRNAs

We selected 11 circRNAs with abnormal expression and verified their sequence data by qRT-PCR. In this paper, the three pairs of samples have been processed as follows. The total RNA of the canine mammary tumour tissues and the adjacent normal tissues was reverse transcribed to synthesize cDNA by using a Primescipt RT reagent kit (TaKaRa, Tokyo, Japan) in accordance with manufacturer’s protocols. The differential expressions of circRNAs were measured with SYBR Green I qPCR SMix (ROX; Roche, Basel, Switzerland) according to the requirements of Roche LightCycler 480 real-time quantitative PCR instrument. The primers of 11 differentially expressed circRNAs that correlated with the occurrence of canine mammary tumour are listed in Table 1. The relative expression of circRNAs was calculated to GPI [37]. Tissue from the tumour portion of the three samples was used as the experimental group, with adjacent normal tissue as the control. Each sample was repeated three times. Each qPCR assay was repeated three times. The 2^−^^ΔΔct^ method was used to calculate the qPCR results.

### 2.8. Statistical Analysis

The results were expressed as the mean ± standard deviation. circRNAs differentially expressed between the two groups were estimated by fold change, and a *t*-test was used for the comparisons among groups. *p*-values of *p* ≤ 0.05 were considered statistically significant.

## 3. Results

### 3.1. Sample Collection and Identification

The sample was made into a paraffin block before being sent for pathological diagnosis, and the LABOKLIN animal clinical laboratory in Germany was commissioned by us for pathological diagnosis. After histopathological analysis, all three tissue samples were malignant [38]. The pathological diagnosis of the sequencing samples were three cases of canine mammary adenocarcinoma. Table 2 describes the case information and the pathological diagnoses. Three cases of canine mammary gland adenocarcinoma were used as three replicates. The cancer tissue was used as the experimental group, and the adjacent normal tissues were used as the control group. B1, D1 and G1 represent the tumour tissues, and B2, D2 and G2 represent the adjacent normal tissues.

### 3.2. Validation of Canine Mammary Tumour Samples’ RNA Quality

The spectrophotometry results showed that the ratio of D (260 nm)/D (280 nm) of the six samples ranged from 1.8 to 2 and that D (260 nm)/D (230 nm) was more than 1.8; the Q30 values were greater than 80%, and agarose gel electrophoresis showed that the total RNA of the 28S and 18S bands were clear and that the 5S bands were blurred. These results indicate that the quality and purity of the RNAs were sufficient for further experimental analysis.

### 3.3. CircRNA Expression Profiles

Canine mammary tumour RNA from three tumour tissues and three adjacent normal tissues were analysed using circRNAs sequencing. In total, the number of clean reads from the three canine mammary tumour tissues were 65,542,390, 68,900,544 and 76,120,034, and the mapped read counts were 46,574,532, 41,665,425 and 53,184,324. The number of clean reads from the three adjacent normal tissues was 74,050,512, 62,169,074 and 83,803,000, and the mapped read counts were 51,944,687, 44,163,639 and 59,550,556. In summary, 14,851 circRNAs were detected by circRNA sequencing, some of which were novel. Among the novel circRNAs in this study, circRNAs derived from exons constituted the majority, and the numbers of circRNAs of each type were not the same (Figure 1A). The number of circRNAs that were between 0 and 500 nt were the most common (Figure 2B). The differentially expressed circRNAs with statistical significance between groups were plotted as volcano plots and scatter plots, as shown in Figure 2A,B.

### 3.4. Differential Expression of CircRNAs in Canine Mammary Tumours

Overall, 106 significantly differentially expressed circRNAs were found in canine mammary tumour tissues compared with adjacent normal tissues. Of these, 64 were upregulated and 42 were downregulated. In addition, seven upregulated and four downregulated circRNAs were related to tumourigenesis; the progression of canine mammary tumours is summarized in Table 3. A hierarchical cluster analysis chart of the 106 differentially expressed circRNAs was generated according to the expression level to account for the distinguishable circRNA expression profiles in the samples (Figure 3). The results showed that the differential expression of circRNAs could correctly distinguish between canine mammary tumour tissues and adjacent normal tissues.

### 3.5. GO and KEGG Pathway Analysis of Differentially Expressed CircRNAs

We performed a GO analysis (Figure 4A) and a KEGG pathway analysis (Figure 4B) on the transcripts that produced circRNAs. The GO analysis showed that most genes were binding proteins, such as catalysis, transmission and other molecular functions. These proteins are associated with biological processes such as regulation of secretory pathways, positive regulation of phagocytosis and regulation of neurotransmitter secretion. The KEGG pathway analysis showed that differentially expressed genes are enriched in the cGMP-PKG signalling pathway, the cAMP signalling pathway and the OXYTOCIN signalling pathway. These genes are often altered in tumour expression. The Pathway analysis revealed that genes such as chr27:26618544-26687235-, chr26:8194880-8201833+ and chr5:40409024-40419360+ were predicted to be involved in canine mammary tumours.

### 3.6. CircRNA-miRNA International Network

Given that circRNAs were found to function as miRNA sponges [39], we assumed that chr5:40409024-40419360+, chr25:42146647-42163429- and chr6:49915037-49927637-may regulate gene expression via a similar mechanism. The molecular interaction of eight significantly differential expressed circRNAs by qRT-PCR and miRNAs are depicted in Figure 5.

### 3.7. CircRNA Signature Classifier Selected by qRT-PCR

To verify the reliability of the circRNA sequencing results, we selected some important genes for three cases of canine mammary tumour tissues and adjacent normal tissues for verification. The qRT-PCR showed that the seven significantly upregulated circRNAs (Figure 6A) as well as the four significantly downregulated circRNAs (Figure 6A) were consistent with this sequencing results.

Fold changes detected by sequencing and qPCR are shown in Figure 6B. The trend of fold change detected by the two methods was consistent, which proved the sequencing results to be highly reliable.

## 4. Discussion

The clinical signs of canine mammary tumours vary over the course of a few days to several months. Most dogs with mammary tumours are clinically healthy when they first present to hospital, but tumours that present clinically within a short period of time are more aggressive than those in dogs with a long clinical history [40]. Common diagnostic methods such as pathology and imaging are limited in the early diagnosis of mammary gland tumours in dogs, and when metastases develop, treatment with the current surgical methods is not curative for the affected dog [41]. Early detection is one of the most important principles in prolonging the life and maintaining the quality of life of infected dogs, so it is worthwhile to explore easy and quick methods of early diagnosis. There is continuing research on the relationship between circRNAs and tumours; many circRNAs have been studied in humans, mice and other species [42,43], and the expression profiles of lung adenocarcinoma, hepatocellular carcinoma and osteosarcoma have been explored [44,45,46]. However, until now, there have been no reports on the abnormal expression of circRNAs in canine mammary tumours. This paper represents the first more complete expression profiling of circRNAs in canine mammary tumours.

circRNAs have recently been identified as disease-related and can act as miRNA sponges to influence parental gene expression [47]. Networks based on sequencing data are a widely accepted method of exploring the function and deregulation of circRNAs as well as the interaction with miRNAs. Chr25:42146647-42163429- is one of several dysregulated circRNAs identified in this study, and the miRNAs with which it interacts are miR-7 and miR-34a. Additionally, miR-7 and miR-34a have been reported to induce breast cancer cell proliferation and migration [48,49]. Thus, this may be our future research direction.

In this study, GO function analysis showed that circRNAs are involved in tumour development and development-related functions, such as promoting cell proliferation, cytoskeletal protein transport, regulating glycoprotein biosynthesis and metabolism, phosphorylation of chondroitin proteoglycan synthesis and metabolism. The KEGG pathway analysis showed that these circRNAs are involved in cancer-related key pathways, including the cAMP pathway. Previous studies have shown that activation of the cAMP pathway can inhibit cell proliferation and can promote cell differentiation [50,51]. The genes of origin of chr27:26618544-26687235- and chr17:7960861-7967766- are involved in the cAMP pathway in the present study and may regulate cell cycle pathways. In this study, a total of 196 circRNAs were identified as part of the cAMP signalling pathway. This pathway plays an important role in the development of breast tumours. Another important pathway is the cGMP PKG signalling pathway, which is involved in the opening of cell membrane ion channels, glycogenolysis and apoptosis [51].

circRNAs are wildly expressed in human and animal tissues and show developmental and stage-specific expression [52]. circRNAs are more stable in mammalian cells compared with miRNAs and lncRNAs [53]. Currently, oncologists have found that nucleolar pressure is a potential antitumour therapy, and circ ANRIL can enrich pre-rRNA and can increase nucleolar pressure [54]. In addition, it has been found that circRNAs exist widely in the blood, saliva and exocrine bodies [55]. These characteristics may enable circRNAs to be an ideal biomarker for animal diseases.

circRNAs also have some drawbacks for diagnosis. First, some circRNAs require the patient’s tissue to be sampled, which can cause trauma to the patient. Second, detecting circRNAs in tissues or exosomes is more expensive than existing assays, limiting the widespread use of circRNAs as biomarkers. In addition, their function may be complex and different from each other due to the large number of circRNAs.

The findings of this study extend the understanding of canine mammary tumours. We found that circRNAs are valuable in the diagnosis of canine mammary tumours. These results have potential clinical implications from the perspective of exploring the biological functions of circRNAs in canine mammary tumours and the molecular mechanisms that exert these functions

## 5. Conclusions

This study represents the first more complete expression profiling of circRNAs in canine mammary tumours. we identified a total of 14,851 circRNAs in canine mammary tumours and its adjacent normal tissues in which 106 were differentially expressed by RNA sequencing. Bioinformatics analysis shows that some genes are closely related to processes of canine mammary tumour development. Additionally, certain genes are involved in tumour-related pathways. These differentially expressed genes are potential targets for the diagnosis and prognosis of canine mammary tumours and establish a foundation for future studies on the pathogenesis and targeted therapy of canine mammary tumours.

## Figures and Tables

**Figure 1 vetsci-09-00205-f001:**
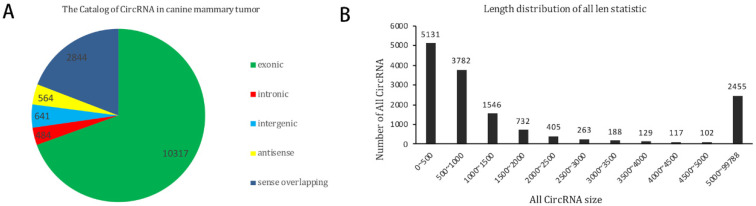
CircRNA sequencing. (**A**) CircRNA number of five catalogues; (**B**) CircRNA number of the predicted sequence length.

**Figure 2 vetsci-09-00205-f002:**
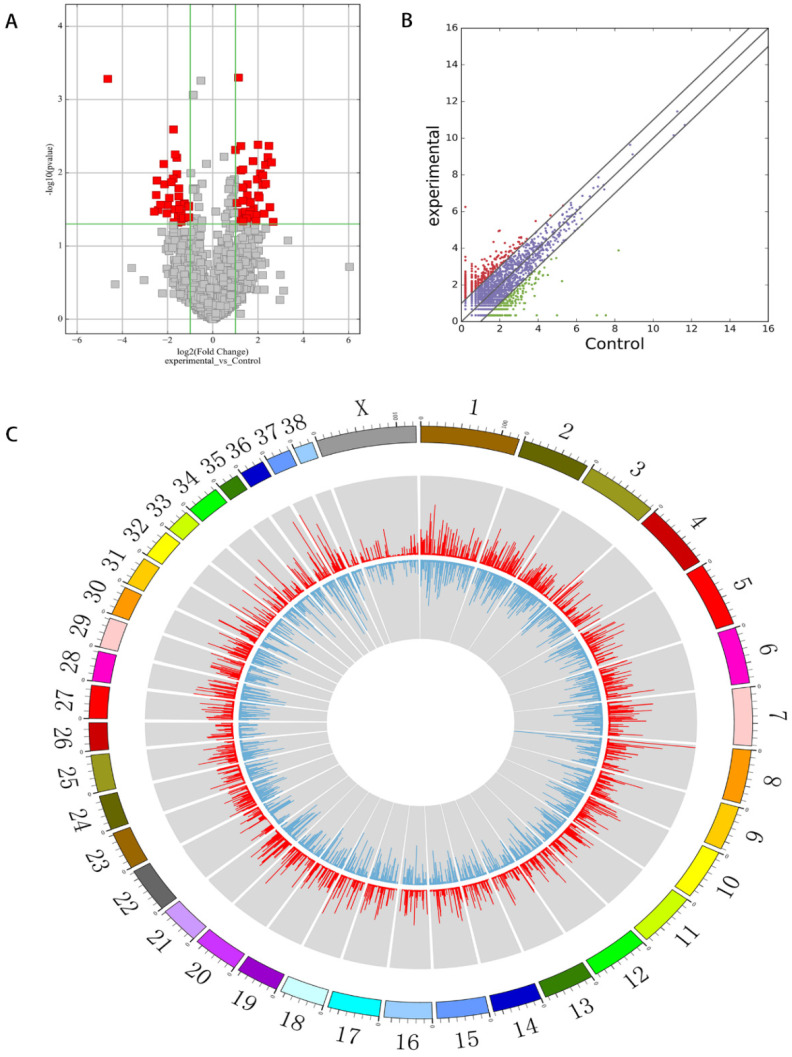
(**A**) Volcano plot filtering. The red area represents differentially expressed circRNAs with fold change > 2, *p* < 0.05; (**B**) scatter plots. Red for high abundance, green for low abundance (fold change > 2, *p* < 0.05); (**C**) circRNA expression circos, outline is the reference genome, inside is the chromosome coverage across all the samples.

**Figure 3 vetsci-09-00205-f003:**
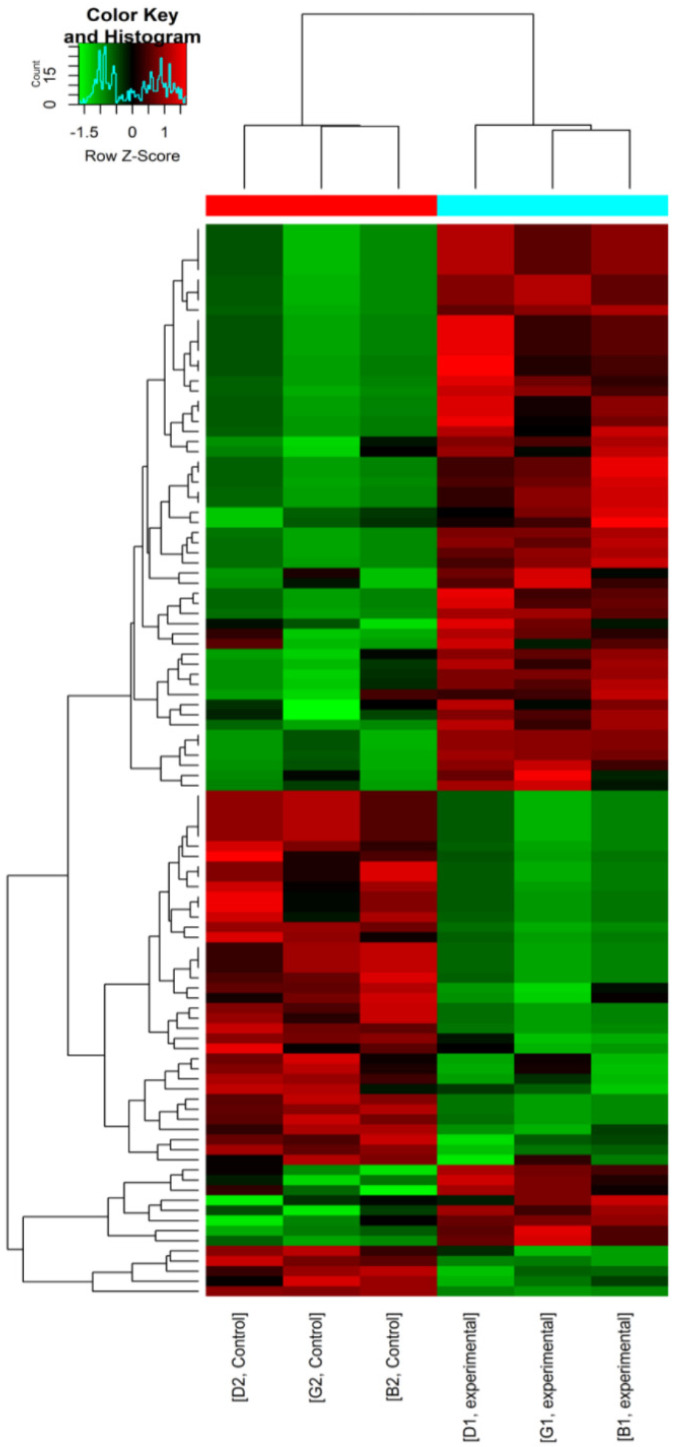
Hierarchical cluster analysis chart of the 106 differentially expressed circRNAs; red represents that the expression level is relatively high, green represents relatively low expression. B1, D1, G1: cancer tissues, B2, D2, G2: adjacent normal tissues.

**Figure 4 vetsci-09-00205-f004:**
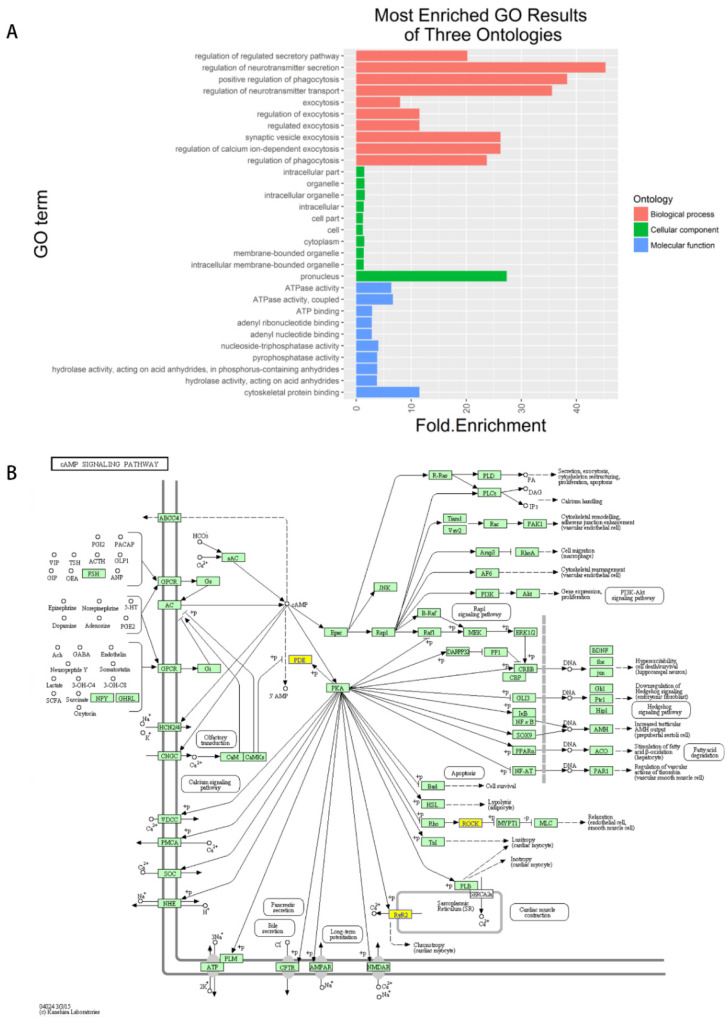
(**A**) Most enriched GO results expression circos. The horizontal axis is the number of significantly different genes, and the vertical axis is the top 10 significantly enriched GO terms. *p* < 0.05 was considered significant. (**B**) The upregulated pathway named cfa04024, where the orange boxes indicate the genes targeted by differentially expressed circRNAs. ^+^: Phosphorylation. ^−^: Dephosphorylation.

**Figure 5 vetsci-09-00205-f005:**
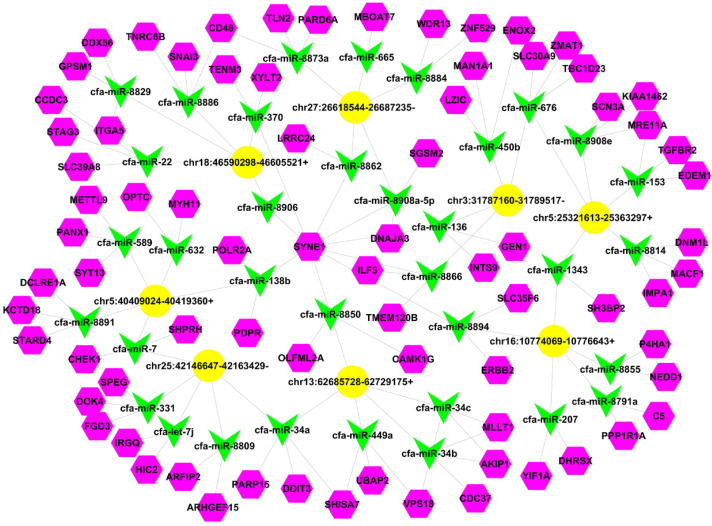
The molecular interaction of eight circRNAs and miRNAs. Yellow represents circRNAs, green represents miRNAs, and purple represents miRNA target genes.

**Figure 6 vetsci-09-00205-f006:**
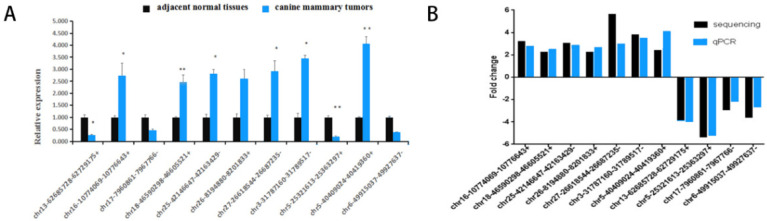
(**A**) Relative expression as indicated by RT-qPCR (11 circRNAs). * *p* < 0.05, ** *p* < 0.01; (**B**) fold change detected using sequencing and qPCR.

**Table 1 vetsci-09-00205-t001:** Primers for qPCR.

CircRNA ID	Primer Sequence (5′-3′)		Product Length (nt)
chr13-62685728-62729175+	F:TGCAACGGTGACAATAGCTC	R:GAGGTCTTCTGTTCCCAAGG	193
chr16-10774069-10776643+	F:ACAAGATTTCTGCCCAGGAG	R:ATTTGGTGGGGATGGGATAG	218
chr17-7960861-7967766-	F:GCTCTCAGAGGAAAGGACGTT	R:TAACCGGGCCGTAGTATCAG	202
chr18-46590298-46605521+	F:CGGAGAGAAGATGCTCACG	R:CTCCCCCAGATGCCTACATA	190
chr25-42146647-42163429-	F:GTTCAGCATCCAGTCCCAGT	R:TGTTATTAGGCCGGTTGGAC	182
chr26-8194880-8201833+	F:AACCCGAAGGAACCTCTGAT	R:CAGGCATTTGCTCGCTCTAT	183
chr27-26618544-26687235-	F:TGATGATGACCCTCACCAAA	R:CATCACGGCAATATCCACAG	183
chr3-31787160-31789517-	F:TATCCCAGTGACGGATGACC	R:GCCTGCTGTTTGGCTAGATT	197
chr5-25321613-25363297+	F:AGTGTCTCCCAGGGTGAATG	R:AATTCCTTTCTGCATCCCTGT	194
chr5-40409024-40419360+	F:TCTGAGCAGCAGAACAAAGC	R:GGAACTGGAATTGGTGCTGT	190
chr6-49915037-49927637-	F:CAGGCCAATATTCCAGCTTC	R:GCTGTCAATAATCCCCAAGC	193
ATCB	F:GGCATCCTGACCCTGAAGTA	R:GGGGTGTTGAAAGTCTCGAA	203

**Table 2 vetsci-09-00205-t002:** The information for patients with canine mammary tumours subjected to a circRNA expression profile sequencing assay.

Patient No.	Age	Gender	Variety	Diagnosis Type
B	15	Sterile females	Pekingese	Mammary gland adenocarcinoma
D	10	Females	Miniature Pinscher	Mammary gland adenocarcinoma
G	13	Sterile females	Beagles	Mammary gland adenocarcinoma

**Table 3 vetsci-09-00205-t003:** The 11 differentially expressed circRNAs that correlated with the occurrence of canine mammary tumours.

CircRNA ID	*p*-Value	Fold Change	Regulation	txStart	txEnd	Original Gene
chr27:26618544-26687235-	0.004298839	5.590525	up	26618543	26687235	*PDE3A*
chr26:8194880-8201833+	0.000503519	2.1959609	up	8194879	8201833	*ATP2A2*
chr3:31787160-31789517-	0.037826954	3.7550404	up	31787159	31789517	*HERC2*
chr18:46590298-46605521+	0.000503519	2.1959609	up	46590297	46605521	*KCNQ1*
chr16:10774069-10776643+	0.025505861	3.1671237	up	10774068	10776643	*CREB3L2*
chr17:7960861-7967766-	0.03811154	2.8914923	down	7960860	7967766	*ROCK2*
chr5:25321613-25363297+	0.032276195	5.3261744	down	25321612	25363297	*GUCY1A2*
chr5:40409024-40419360+	0.009322253	2.3655272	up	40409023	40419360	*ULK2*
chr25:42146647-42163429-	0.044174822	3.0292449	up	42146646	42163429	*TRIP12*
chr6:49915037-49927637-	0.013468932	3.5655166	down	49915036	49927637	*MFSD14A*
chr13:62685728-62729175+	0.022070076	3.7889268	down	62685727	62729175	*MTHFD2L*

## Data Availability

The dataset for this study can be found in NCBI’s Gene Expression Omnibus and can be accessed via GEO series accession number GSE137825 (https://www.ncbi.nlm.nih.gov/geo/query/acc.cgi?acc=GSE137825 (accessed on 29 November 2021)).

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
