# Peer review of "CircRNA Expression Profiles in Canine Mammary Tumours"

_vetsci, 2022, doi:10.3390/vetsci9050205_

Round 1
Reviewer 1 Report
In this study, the authors sought to show the expression profile of circRNAs in canine mammary tumors and, by RNA sequencing and qPCR, identified circRNAs that are characteristically expressed in canine mammary tumors. The analysis suggested that some genes are involved in the progress of canine mammary tumors and tumor-associated pathways. With these results, the authors attempted to establish a basis for future research on new diagnosis and treatment of canine mammary tumors. However, I have some comments, which I believe will improve the readability of the paper.
1. Name of circRNAs The authors should explain how to decide the name of circRNAs by appropriate references, including the meaning of the “+" and “-" symbols? 2. Sequence of circRNAs identified in this study The authors should show the sequences of circRNAs identified in this study. Alternatively, when the sequences have been deposited in the database, the accession numbers should be shown. 3. Nomalization of qPCR results The Ct values for housekeeping transcripts are used to normalize the Ct values for each circRNA. Please describe the name of housekeeping genes used in this study. Minor points L126 I can not find “)” in the sentence. L173 "28 s, 18 s Bands” should be ”28S, 18S bands"
Author Response
Thank you very much for your interest in our paper and for giving us your valuable comments and opinions. We have revised the manuscript in accordance with your suggestions. We sincerely hope that this manuscript will eventually be accepted. Thank you very much for your help and we look forward to hearing from you.
Referee 1
Thank you again for your comments and the following is a detailed reply.
Point 1: Name of circRNAs The authors should explain how to decide the name of circRNAs by appropriate references, including the meaning of the “+" and “-" symbols?
Response 1:Here circRNAs are named by the sequencing company and represent the coordinate information of its genes. In the table of cloud sequenced organisms, the positive and negative strand information is appended to the gene coordinates with a "+" and a "-".
Point 2: Sequence of circRNAs identified in this study the authors should show the sequences of circRNAs identified in this study. Alternatively, when the sequences have been deposited in the database, the accession numbers should be shown.
Response 2:The dataset for this study can be found in NCBI's Gene Expression Omnibus and can be accessed via GEO series accession number GSE137825 ( https://www.ncbi.nlm.nih.gov/geo/query/acc.cgi?acc=GSE137825 ).
Point 3: Nomalization of qPCR results The Ct values for housekeeping transcripts are used to normalize the Ct values for each circRNA. Please describe the name of housekeeping genes used in this study. Minor points L126 I can not find “)” in the sentence. L173 "28 s, 18 s Bands” should be ”28S, 18S bands" .
Response 3: The housekeeping gene used in this study was ACTB. We have modified it according to your comments
The revised draft has been uploaded to the system.
Reviewer 2 Report
In the methods the animal ethics is not cited for the tissue collection. Please ensure correct animal ethics permits are cited for the jurisdiction. To collect normal tissue from a patient as well as cancer tissue should require animal ethics.
Please include the WHO classification of the tumors selected in this study (83. Misdorp W, Else R, Hellmen E, Lipscomb T. Histological Classification of the Mammary Tumors of the Dog and the Cat. World Health Organization International Histological Classification of Tumors of Domestic Animals second series. Silver Spring, MD: American Registry of Pathology; (1999) 7:1–59.) and the staging information of the patients ie local and metastatic disease.
The resolution of some figures and images needs improvement. Especially figure 4 and 6.
Discussion. First sentence. The type and malignancy of canine mammary cancers can vary geographically. In animals with high neutering in the population the incidence can be reduced, and many tumors are benign mixed morphology. So I don't agree with this statement. Further typically early diagnosis and treatment is generally not an issue. I would recommend reviewing this reference and including or changing this introduction to be more specific. https://www.ncbi.nlm.nih.gov/pmc/articles/PMC7198768/
Perhaps just focus on the molecular diagnostics in the discussion.
Author Response
Thank you very much for your interest in our paper and for giving us your valuable comments and opinions. We have revised the manuscript in accordance with your suggestions. A detailed response is attached. We sincerely hope that this manuscript will eventually be accepted. Thank you very much for your help and we look forward to hearing from you.
Referee 2
Thank you again for your comments and the following is a detailed reply.
Point 1: In the methods the animal ethics is not cited for the tissue collection. Please ensure correct animal ethics permits are cited for the jurisdiction. To collect normal tissue from a patient as well as cancer tissue should require animal ethics.
Response1:The ethical clearance document was submitted on the website when the manuscript was originally submitted. we apologize for this omission as the corresponding text description was not included in our paper. Please see the other attachments for the document of animal ethics permits and the corresponding text description has been added to the revised manuscript.
Point 2:Please include the WHO classification of the tumors selected in this study (83. Misdorp W, Else R, Hellmen E, Lipscomb T. Histological Classification of the Mammary Tumors of the Dog and the Cat. World Health Organization International Histological Classification of Tumors of Domestic Animals second series. Silver Spring, MD: American Registry of Pathology; (1999) 7:1–59.) and the staging information of the patients ie local and metastatic disease.
Response2:Please see attached for all pathology report information.
Point 3:The resolution of some figures and images needs improvement. Especially figure 4 and 6.
Response 3:Revisions have been made in response to your comments.
Point 4:Discussion. First sentence. The type and malignancy of canine mammary cancers can vary geographically. In animals with high neutering in the population the incidence can be reduced, and many tumors are benign mixed morphology. So I don't agree with this statement. Further typically early diagnosis and treatment is generally not an issue. I would recommend reviewing this reference and including or changing this introduction to be more specific.
https://www.ncbi.nlm.nih.gov/pmc/articles/PMC7198768/
Perhaps just focus on the molecular diagnostics in the discussion
Response 4:Revisions have been made in response to your comments.
The revised draft has been uploaded to the system.
Reviewer 3 Report
The manuscript presents the results of a study aimed at describing the circRNAs expression profile in three pairs of canine mammary tumors and the adjacent non-neoplastic tissue from the same patients using NGS and confirmed by qRT-PCR. Upregulated and downregulated circRNAs were identified and the main regulatory functions involved were highlighted by gene ontology and KEGG pathway analysis. The introduction is consistent with the state of the art of molecular oncologic diagnostics and tumor biology, but the bibliography could be updated. The transcriptome and non-coding RNAs represent undoubtedly pivotal research topics in human and canine tumor growth and further studies in this field are needed. The methodology is clearly described, and the English language and style are fine, even if a revision of a native English speaker could improve the text quality. In conclusion, I believe that this manuscript could be accepted for publication after some minor revision.
Minor revisions:
Page 1 line 11. As outlined in the vetsci words layout, the authors should cancel the headings “Background/Aims:; Methods:; Results:; Conclusion:” from the abstract.
Page 1 line 15. The authors should change “our study” to “this study”, avoiding the use of the first person. They should also check it in the rest of the manuscript.
Page 1 line 33. The authors should add “to develop” after “…screening dogs with high-risk…”
Page 1 line 58. The authors should add some information regarding circRNAs and miRNAs studies in veterinary medicine or at least domestic animals cell cultures.
Page 2 line 49. The authors should add more recent references regarding circRNA in cancer.
Page 2 line 81. How many samples were collected? If the sampling involved only the three neoplastic samples (+ three control samples) described afterwards, please add here the sampling size.
Page 3 line 126. Check the URL.
Page 3 lines 130-134. The authors should move these lines in the introduction.
Page 4 line 148. How many samples were submitted to NGS before? If all the NGS-tested samples were also tested by qRT-PCR analysis, please rephrase this sentence.
Page 4 line 159. Regarding the pathological diagnosis, was also collected FFPE tissue, or a part of the 2cm frozen sample was sent to LABOKLIN?
Page 4 line 162. Please switch Table 2 and Table 3.
Page 4 lines 167-168. Please cancel the redundant sentence. Do the authors think that using adjacent healthy tissue from neoplastic-developing subjects as a negative control, compared to tissue from non-neoplastic subjects, could represent a limitation of this study?
Page 4 line 193. The authors should better clarify if the 11 circRNAs were associated with tumourigenesis after GO and KEGG pathway analysis or was previously reported by other researchers. May add the references.
Page 5 line 194. Please switch Table 2 and Table 3.
Page 6 line 229. Given the importance of the reproductive status in CMT pathogenesis the author should add if the subject were spayed or not; and if so, at what age they were neutered.
Page 10 line 263. Please add some references.
Page 10 line 264. Please rephrase the sentence “Although some scholars in our country…” avoiding the use of the first person and country landmark. Please, add also some references.
Page 11 line 280. Please rephrase the sentence avoiding the first person possessive.
Page 11 line 293. The authors should better highlight the study’s limitations and weaknesses.
Author Response
Thank you very much for your interest in our paper and for giving us your valuable comments and opinions. We have revised the manuscript in accordance with your suggestions. We sincerely hope that this manuscript will eventually be accepted. Thank you very much for your help and we look forward to hearing from you.
Referee 3
Thank you again for your comments and the following is a detailed reply.
Point 1: Page 1 line 11. As outlined in the vetsci words layout, the authors should cancel the headings “Background/Aims:; Methods:; Results:; Conclusion:” from the abstract.
Response 1: Revisions have been made in response to your comments.
Point 2: Page 1 line 15. The authors should change “our study” to “this study”, avoiding the use of the first person. They should also check it in the rest of the manuscript.
Response 2: Revisions have been made in response to your comments.
Point 3: Page 1 line 33. The authors should add “to develop” after “…screening dogs with high-risk…”
Response 3: Revisions have been made in response to your comments.
Point 4: Page 1 line 58. The authors should add some information regarding circRNAs and miRNAs studies in veterinary medicine or at least domestic animals cell cultures.
Response 4: Revisions have been made in response to your comments.
Point 5: Page 2 line 49. The authors should add more recent references regarding circRNA in cancer.
Response 5: Revisions have been made in response to your comments.
Point 6: Page 2 line 81. How many samples were collected? If the sampling involved only the three neoplastic samples (+ three control samples) described afterwards, please add here the sampling size.
Response 6: A total of 17 samples were collected and three cases of malignant breast tumours were finally selected for follow-up experiments.
Point 7: Page 3 line 126. Check the URL.
Response 7: Revisions have been made in response to your comments.
Point 8: Page 3 lines 130-134. The authors should move these lines in the introduction.
Response 8: Revisions have been made in response to your comments.
Point 9: Page 4 line 148. How many samples were submitted to NGS before? If all the NGS-tested samples were also tested by qRT-PCR analysis, please rephrase this sentence.
Response 9: We submitted three samples to NGS and performed qRT-PCR on these three samples.
Point 10:Page 4 line 159. Regarding the pathological diagnosis, was also collected FFPE tissue, or a part of the 2cm frozen sample was sent to LABOKLIN?
Response 10: The sample was made into a paraffin block before being sent for pathological diagnosis. The corresponding text description has been added to the revised version.
Point 11:Page 4 line 162. Please switch Table 2 and Table 3.
Response 11: Revisions have been made in response to your comments.
Point 12:Page 4 lines 167-168. Please cancel the redundant sentence. Do the authors think that using adjacent healthy tissue from neoplastic-developing subjects as a negative control, compared to tissue from non-neoplastic subjects, could represent a limitation of this study?
Response 12: Revisions have been made in response to your comments.
Point 13:Page 4 line 193. The authors should better clarify if the 11 circRNAs were associated with tumourigenesis after GO and KEGG pathway analysis or was previously reported by other researchers. May add the references.
Response 13: The 11 circRNAs selected for this study were from the 106 differentially expressed circRNAs in the predicted results. After qRT-PCR, seven of these 11 circRNAs were highly expressed in the tumour tissue, while four were lowly expressed (Figure 6.). Unfortunately, due to the limitations of this study, no functional validation was performed.
Point 14:Page 5 line 194. Please switch Table 2 and Table 3.
Response 14: Revisions have been made in response to your comments.
Point 15:Page 6 line 229. Given the importance of the reproductive status in CMT pathogenesis the author should add if the subject were spayed or not; and if so, at what age they were neutered.
Response 14: Information on whether the dog was spayed or neutered has been added to the revision. Upon our consultation, the affected dog was neutered prior to adoption and the owner did not know the exact age of neutering.
Point 15:Page 10 line 263. Please add some references.
Response 15: Revisions have been made in response to your comments.
Point 16:Page 10 line 264. Please rephrase the sentence “Although some scholars in our country…” avoiding the use of the first person and country landmark. Please, add also some references.
Response 16: Revisions have been made in response to your comments
Point 17:Page 11 line 280. Please rephrase the sentence avoiding the first person possessive.
Response 17: Revisions have been made in response to your comments.
Point 18:Page 11 line 293. The authors should better highlight the study’s limitations and weaknesses.
Response 18: Revisions have been made in response to your comments.
The revised draft has been uploaded to the system.
Round 2
Reviewer 1 Report
Thank you for your response to my comments. I have additional comments to you. To Response 1 Because the IDs from the sequence company could be tentative, I recommend you to consider giving appropriate names to circRNAs like “hsa_circ_0000745” refereed in the introduction. It could be possible that the IDs based on the nucleotide number may change when the genome assembly version changes. I also recommend you to add a supplemental scheme showing the mechanisms of circRNA formation regarding the positive and negative strand information with "+" and a “-“ symbols. To Response 2 I was unable to get the sequence data in the file “GSE137825_CircRNA_Expression_Profiling.xlsx” from the entry you kindly shown to me. The information I have got for “chr27:26618544-26687235-“ is uploaded as an example. Because it may be difficult for the readers to know the sequence, could you explain in the paper where and how the sequence can be obtained. Alternatively, please consider to show the sequence or the txStart and txEnd in supplementary materials. To Response 3 I recommend you to show the data for the expression of ACTB in this study, because ACTB is closely associated with a variety of cancers and accumulating evidence indicates that ACTB is de-regulated. DOIs: 10.1016/j.cca.2012.12.012, 10.1080/21655979.2021.1973220
Author Response
Thank you very much for your comments and suggestions, they will be of great help to us in improving the quality of our work in the future.
Referee 1
Thank you again for your comments and the following is a detailed reply.
Point 1: Because the IDs from the sequence company could be tentative, I recommend you to consider giving appropriate names to circRNAs like “hsa_circ_0000745” refereed in the introduction. It could be possible that the IDs based on the nucleotide number may change when the genome assembly version changes. I also recommend you to add a supplemental scheme showing the mechanisms of circRNA formation regarding the positive and negative strand information with "+" and a “-” symbols.
Response 1: Thank you very much for your suggestion. In fact, we do not intend to change the name currently in order to facilitate the search in the file "GSE137825_CircRNA_Expression_Profiling.xlsx". The "+" and "-" in the manuscript are already determined by the coordinates of the most suitable transcript for the CircRNAs.
Point 2: To Response 2 I was unable to get the sequence data in the file “GSE137825_CircRNA_Expression_Profiling.xlsx” from the entry you kindly shown to me. The information I have got for “chr27:26618544-26687235-“ is uploaded as an example. Because it may be difficult for the readers to know the sequence, could you explain in the paper where and how the sequence can be obtained. Alternatively, please consider to show the sequence or the txStart and txEnd in supplementary materials.
Response 2: We apologize for this confusion. Please use the information under the column U in the file "GSE137825_CircRNA_Expression_Profiling.xlsx" and search in another database (http://asia.ensembl.org/) to obtain circRNA information.
Point 3: To Response 3 I recommend you to show the data for the expression of ACTB in this study, because ACTB is closely associated with a variety of cancers and accumulating evidence indicates that ACTB is de-regulated. DOIs: 10.1016/j.cca.2012.12.012, 10.1080/21655979.2021.1973220
Response 3: In two recently published articles (â‘ Cacciola NA, Sgadari M, Sepe F, et al. Metabolic Flexibility in Canine Mammary Tumors: Implications of the Carnitine System. Animals (Basel). 2021;11(10):2969. Published 2021 Oct 15. doi:10.3390/ani11102969;②Ou G, Jiang X, Gao A, Li X, Lin Z, Pei S. Celastrol Inhibits Canine Mammary Tumor Cells by Inducing Apoptosis via the Caspase Pathway. Front Vet Sci. 2022;8:801407. Published 2022 Feb 4. doi:10.3389/fvets.2021.801407), ACTB was used as an internal reference gene for western blot analysis and qRT-PCR analysis, so we believe that the use of ACTB as an internal reference gene has been feasible so far in canine mammary tumour studies. However, in human clinical studies, ACTB was deregulated in different tumor invasiveness degrees and TNM stages of hepatocellular carcinoma and 3’-UTR of ACTB has been demonstrated to play an important role in the process of HCC development. In the future, whether to use ACTB alone as an internal reference gene is a question to be pondered and we will follow this topic.
Referee 1
Thank you again for your comments and the following is a detailed reply.
Point 1: Because the IDs from the sequence company could be tentative, I recommend you to consider giving appropriate names to circRNAs like “hsa_circ_0000745” refereed in the introduction. It could be possible that the IDs based on the nucleotide number may change when the genome assembly version changes. I also recommend you to add a supplemental scheme showing the mechanisms of circRNA formation regarding the positive and negative strand information with "+" and a “-” symbols.
Response 1: Thank you very much for your suggestion. In fact, we do not intend to change the name currently in order to facilitate the search in the file "GSE137825_CircRNA_Expression_Profiling.xlsx". The "+" and "-" in the manuscript are already determined by the coordinates of the most suitable transcript for the CircRNAs.
Point 2: To Response 2 I was unable to get the sequence data in the file “GSE137825_CircRNA_Expression_Profiling.xlsx” from the entry you kindly shown to me. The information I have got for “chr27:26618544-26687235-“ is uploaded as an example. Because it may be difficult for the readers to know the sequence, could you explain in the paper where and how the sequence can be obtained. Alternatively, please consider to show the sequence or the txStart and txEnd in supplementary materials.
Response 2: We apologize for this confusion. Please use the information under the column U in the file "GSE137825_CircRNA_Expression_Profiling.xlsx" and search in another database (http://asia.ensembl.org/) to obtain circRNA information.
Point 3: To Response 3 I recommend you to show the data for the expression of ACTB in this study, because ACTB is closely associated with a variety of cancers and accumulating evidence indicates that ACTB is de-regulated. DOIs: 10.1016/j.cca.2012.12.012, 10.1080/21655979.2021.1973220
Response 3: In two recently published articles (â‘ Cacciola NA, Sgadari M, Sepe F, et al. Metabolic Flexibility in Canine Mammary Tumors: Implications of the Carnitine System. Animals (Basel). 2021;11(10):2969. Published 2021 Oct 15. doi:10.3390/ani11102969;②Ou G, Jiang X, Gao A, Li X, Lin Z, Pei S. Celastrol Inhibits Canine Mammary Tumor Cells by Inducing Apoptosis via the Caspase Pathway. Front Vet Sci. 2022;8:801407. Published 2022 Feb 4. doi:10.3389/fvets.2021.801407), ACTB was used as an internal reference gene for western blot analysis and qRT-PCR analysis, so we believe that the use of ACTB as an internal reference gene has been feasible so far in canine mammary tumour studies. However, in human clinical studies, ACTB was deregulated in different tumor invasiveness degrees and TNM stages of hepatocellular carcinoma and 3’-UTR of ACTB has been demonstrated to play an important role in the process of HCC development. In the future, whether to use ACTB alone as an internal reference gene is a question to be pondered and we will follow this topic.